# A Zero-Waste Multi-Criteria Decision-Support Model for the Iron and Steel Industry in Developing Countries: A Case Study

**Yolandi Schoeman** [1,*] , **Paul Oberholster** [1] **and Vernon Somerset** [2]

1   Centre for Environmental Management, Faculty of Natural and Agricultural Sciences, University of the Free State, Bloemfontein 9301, South Africa; oberholsterpj@ufs.ac.za
2   Chemistry Department, Faculty of Applied Sciences, Cape Peninsula University of Technology, Bellville 7535, South Africa; somersetv@cput.ac.za
*   Correspondence: schoeman.yolandy@gmail.com

**Abstract:** The iron and steel industry is a major global industry that consumes vast quantities of energy and causes environmental degradation through greenhouse gas emissions and industrial waste generation, treatment, and disposal. There is a need to manage complex iron and steel industrial waste in Africa, which requires a system engineering approach to zero waste management as informed by multi-criteria decision-making. The purpose of the current study was to develop a hybrid four-step multi-criteria decision-support model, the i-ZEWATA (Industrial Zero Waste Tiered Analysis). I-ZEWATA acts as a road map to understand, design, assess, and evaluate the iron and steel industrial waste systems with the ultimate objective of moving towards and achieving a zero-waste footprint. The results demonstrate that iron and steel waste can be identified, visualized, prioritized, and managed to promote zero-waste by applying a system-engineered approach. Additionally, relationship patterns to environmental, social, operational, and economic aspects with system behavioral patterns and outcomes were identified. It was clear from the case study in South Africa that, although technology and solution investment is essential, waste management, valorization, and treatment components require a concerted effort to improve industrial waste operational management through effective zero-waste decision-support towards a circular economy.

**Keywords:** zero-waste; industrial waste management; circular economy; hazardous waste management; multi-criteria decision-support; i-zewata

## 1. Introduction

The iron and steel industry is a major global industry consuming vast quantities of energy and causing environmental degradation through greenhouse gas emissions and industrial waste generation and disposal. A significant concern is the generation of significant industrial waste volumes, such as slag, as part of the manufacturing process [1]. In 2017 more than 400 million tons of slag at a density ranging between 2.5 to 3.5 t/m$^3$ [2] was produced globally [3]. Even though steel is referred to as the world's most recyclable material, material flows within the industry have been demonstrated to be environmentally unsustainable and creating environmental burdens throughout the material and process life cycles [4,5].

An outcome of improper iron and steel industrial waste management can result in releases of toxic contaminants as water, soil, and air pollution from disposal facilities and uncontrolled waste discharges. Toxic contaminant releases encountered in industrial waste management and treatment systems, such as operational activities, including collection, storage, and transportation of waste [6–8], contribute to economic and environmental losses and socio-economic strains [9]. Contaminant releases also hinder economic development and ecosystem sustainability. One can now argue that complications resulting from toxic contaminant releases are not only limited to direct waste discharges or inefficient disposal practices but result from operational decisions themselves.

The perceived view of industrial waste as a burden is contrasted by the view that managing waste [10] makes business sense. When industrial waste's integrated and systems-based nature is understood, improvements to waste management systems unlock value in waste, further enabling industrial waste opportunities. Consequently, industrial waste opportunities exist through waste reduction, reuse, and minimization in the manufacturing process by substituting virgin materials. Additionally, industries can participate in waste exchange and -trade initiatives through industrial symbiosis programs where waste can be used as input material. Existing and historic waste streams can be developed to create viable economic and social development opportunities, promote industrial symbiosis, and contribute towards a circular economy. Hence, what is absent from the current industrial waste management approach at iron and steel facilities is a systems-based decision-support model to inform operational decisions guided by a circular integrated waste management (CIWM) approach.

When CIWM guides industrial waste management systems in a circular economy context, environmental impacts are reduced (such as global warming), resource exchange networks through industrial symbiosis are established, industrial resilience promoted [11], and an improvement in energy and resource efficiencies are achieved [12–14]. In contrast to conventional linear industrial waste management approaches, CIWM aims to transform waste into a resource to reverse the dominant linear trend of processing and consumption and the utilization of raw materials. In a CIWM system, more focus is placed on waste valorization than waste treatment in industrial waste management systems so that waste can be transformed into products that are capable of providing society with a valuable service [12–14]. The availability of sustainable consumption and production (SCP) tools to support circular economy initiatives such as Environmental Management Systems (EMS), Environmental Management Accounting (EMA), the Eco-Management and Audit Scheme (EMAS), International Organization for Standardization (ISO) standards and certification, Eco-Label labeling schemes, carbon footprint reduction programs, Environmental Technology Verification (ETV) for example, can play a critical role in enabling circular economy initiatives and measure implementation progress in industrial waste management [15–24]. However, by implementing existing SCP tools, it does not necessarily imply that industries are managing waste in the interest of promoting a circular economy [15,25–31]. The reason for this limitation can be ascribed to applying methodology that contributes to an inefficient understanding of complex industrial waste systems where the outcome cannot be predicted due to inherent levels of uncertainty [12] that exist within the system. In industrial waste management systems, waste composition is the paramount variable that creates uncertainty and can further cause a dysfunctional configuration of industrial waste management and restrict circular economy initiatives [12,32].

Limited research is available on developing decision-support models to design CIWM for industrial wastes informed by a system engineered approach [33], specifically towards zero waste goal and CIWM strategy achievement [34], as a circular economy enabler [12]. Due to the limitations of linear waste management [13] and the consequent increase in environmental pressure resulting from an amalgamation of stressors [35], applying a system engineered approach can indeed support CIWM. Due to the inherently complicated issues associated with industrial waste management [36], waste composition [13], and the critical requirement to choose the right option for waste valorization, disposal [37], and management, researchers have focused their attention on Multiple Criteria Decision-Making (MCDM) processes as a system engineering method [38] in operational research [39] rather than on Life Cycle Assessment (LCA) [40–42]. As LCA's are regarded as a linear steady-state model of physical flows [43,44] mainly applied to quantifying environmental impacts in waste management, the tool has limitations when applied to dynamic industrial waste systems in CIWM [40,42,45].

It remains challenging to select the most suitable criteria portfolio among various alternatives as the decision-making process is a typical MCDM problem that must consider multiple conflicting criteria [34,46]. In solid waste management, [47] confirmed that

Multiple Criteria Decision-Analysis (MCDA) methods are among the most comprehensive and useful decision support frameworks for decision-support in solid waste management problems and challenges. The strength of the MCDA methods is confirmed by handling several criteria and evaluating alternatives using different methods [48]. Thus, decision analysis tools are essential to support industrial waste management practices [38]. MCDM methods are therefore regarded as a viable method to address waste management in the iron and steel industry.

However, published waste management studies in the iron and steel industry in developing countries, especially in Africa, are limited, especially in promoting CIWM. Published studies in developing countries demonstrate a clear gap in applying a system engineered approach to industrial waste management [9,49–53]. In India, waste from steel plants is minimized, and recycling efforts are maximized based on a management approach informed by economic pressure and the implementation requirements of stricter environmental legislation [48,54,55]. A growing number of projects are initiated in the zero-waste field in developing countries' iron and steel industries. Projects include Arcelor Mittal Tubarao in Brazil utilizing steelmaking slag for road construction and Usiminas in Brazil, paving rural roads with steel co-products [3].

In South Africa, which acts as a case study area for the current study, the existing industrial waste management approach is (1) capital intensive; (2) is indicative of inadequately designed and implemented industrial waste management systems, and (3) waste treatment systems; (4) underlines a lack of understanding of the nature of industrial waste; (5) lacks appropriate methodology to guide industrial waste management and (6) treatment systems development; (7) implements, assesses and treats waste as a non-value adding component to the triple bottom line and is (8) limited in suggesting appropriate SCP tools to promote CIWM [56]. The reason for undertaking the current study was to develop a zero-waste (ZW) multi-criteria decision-support model for the iron and steel industry that will guide waste management, treatment, and valorization towards system engineering and assessment, design, implementation, and evaluation. The developed ZW multi-criteria decision-support model aims to promote waste valorization as a value-adding component to the triple bottom line, to guide the iron and steel industry towards developing CIWM systems, and therefore enabling a circular economy.

The study proposes a hybrid ZW multi-criteria decision support model, i-ZEWATA (Industrial Zero Waste Tiered Analysis), to develop a ZW model and implement a system engineered approach to waste management and valorization for the iron and steel industry with the ultimate objective of achieving ZW. According to the authors' knowledge, this is the first study to develop a ZW multi-criteria decision-support model for the iron and steel industry for developing countries in Africa that can additionally be applied as an SCP tool to promote CIWM.

The structure of the manuscript is as follows: first, the i-ZEWATA model and methodology are described. Second, the methodology as applied to the case study is illustrated, and the results presented. Finally, conclusions are drawn from the study's findings, emphasizing the model's application as an SCP tool to promote CIWM.

## 2. Materials and Methods

### 2.1. The i-ZEWATA Model and Methodology

Although there are a large number of MCDA methods available, no particular method is perfect for applying in every decision-making situation to solve every decision problem [57,58]. Differences in decision recommendations and different results can be an outcome when applying different MCDA methods to decision-making challenges [58–60]. When addressing a specific decision-making challenge, selecting the MCDA method needs to be appropriate to the specific problem that needs to be solved [58,61]. The most applied decision-making (DM) techniques as part of MCDM applied in waste management include complete aggregation models such as the "Technique for Order Preference with Similarity to Ideal Solution" (TOPSIS) and "Choosing by Advantages" (CBA). Partial aggregation

techniques such as AHP, and Fuzzy Logic Decision Making [62] also comprise the most applied DM techniques [62,63] in waste management. Following the completion of an extensive literature review, AHP (including ANP and fuzzy AHP) was found to be the most dominant MCDA tool used, "Preference Ranking Organization Method for Enrichment Evaluations" (PROMETHEE) was an emerging method from the studies, "ELimination Et Choix Traduisant la REalité" (ELECTRE) was the most consistently used method whilst the diversity of MCDA methods was also expanding to include new methods [64]. The application of the MCDA methods "VlseKriterijumska Optimizacija I Kompromisno Resenje" (VIKOR) [58,65] and "Complex Proportional Assessment" (COPRAS) in waste management was found not to be as extensively applied in waste management challenges as AHP. A possible explanation relates to the proper determination of criteria weights and how the various MCDA methods are applied (whether as a singular or as a combination method) that affects the final ranking [58,66,67]. As a new method, "the characteristic objects method" (COMET) [68,69] also has merit in the future application in industrial waste management due to factors associated with the ease of identifying non-linear and linear decision-making and the independence of assessed alternative sets to the assessed decision variants [68,69].

AHP was found to be the most widely used and popular method to apply to a variety of problems that require complex decision-making [64] and for determining weights of criteria in MCDA [70,71]. The growth of publications related to the AHP method stands out above all other techniques [72]. The significance of AHP is based on the principle that to make a decision, people's knowledge and experience are considered as valuable as the data used [62]. It can deal with intangible and or quantifiable data and is carried out in two phases: hierarchy design and evaluation [62]. Experience and knowledge of the problem area are required to design the hierarchies [73]. The application of the ANP methodology is also documented widely, although only recently in the past two decades, with application in strategic policy planning [74,75]; in civil engineering [76]; territorial and environmental assessment [77–82]; and manufacturing systems [83,84]. The ANP, as a multi-criteria methodology, uses a complex model to consider a wide range of qualitative and quantitative criteria [85]. The ANP methodology addresses many decision problems that cannot be structured hierarchically, where many decision problems imply interactions and dependence between the highest and lowest elements [86]. In a hierarchy, the importance of the criteria causes the importance of alternatives, and the importance of the alternatives causes the importance of the criteria [86]. Therefore, by the generalized approach of the super-matrices introduced by the AHP, the ANP extends the applications of the AHP to the cases of interdependent relationships between the assessment elements [87] and can also address the complications associated with rank reversals. Even though rank reversals are as part of decision making as rank preservation, rank can be preserved by using the ideal mode in AHP in both relative and absolute measurement [88–90] and possible future integration of the COMET method and supporting methodology needs to be investigated [91].

The proposed i-ZEWATA model (Figure 1) addresses the waste management system, and valorization practices (that include waste treatment) among a set of predetermined industrial waste management alternatives, namely best practice, business-as-usual, ZW, and the compliance approach [92] aligned to the South African context. A hybrid combined Analytic Hierarchy Process (AHP) and Analytic Network Process (ANP), as a decision support approach, was applied based on the motivational support of [72], namely: (1) that the method allows for the analysis of complex industrial waste decision-making problem by using a systematic approach that breaks down the main problem into practical and simpler sub-problems, (2) ANP is used where there are interdependencies among a group of elements (criteria and alternatives); (3) the detailed analysis of interdependencies and priorities between clusters' elements forces the careful reflection specifically on the decision-making problem that will result in a reliable final decision and better knowledge of the problem. The i-ZEWATA model methodology consisted of the following three steps.

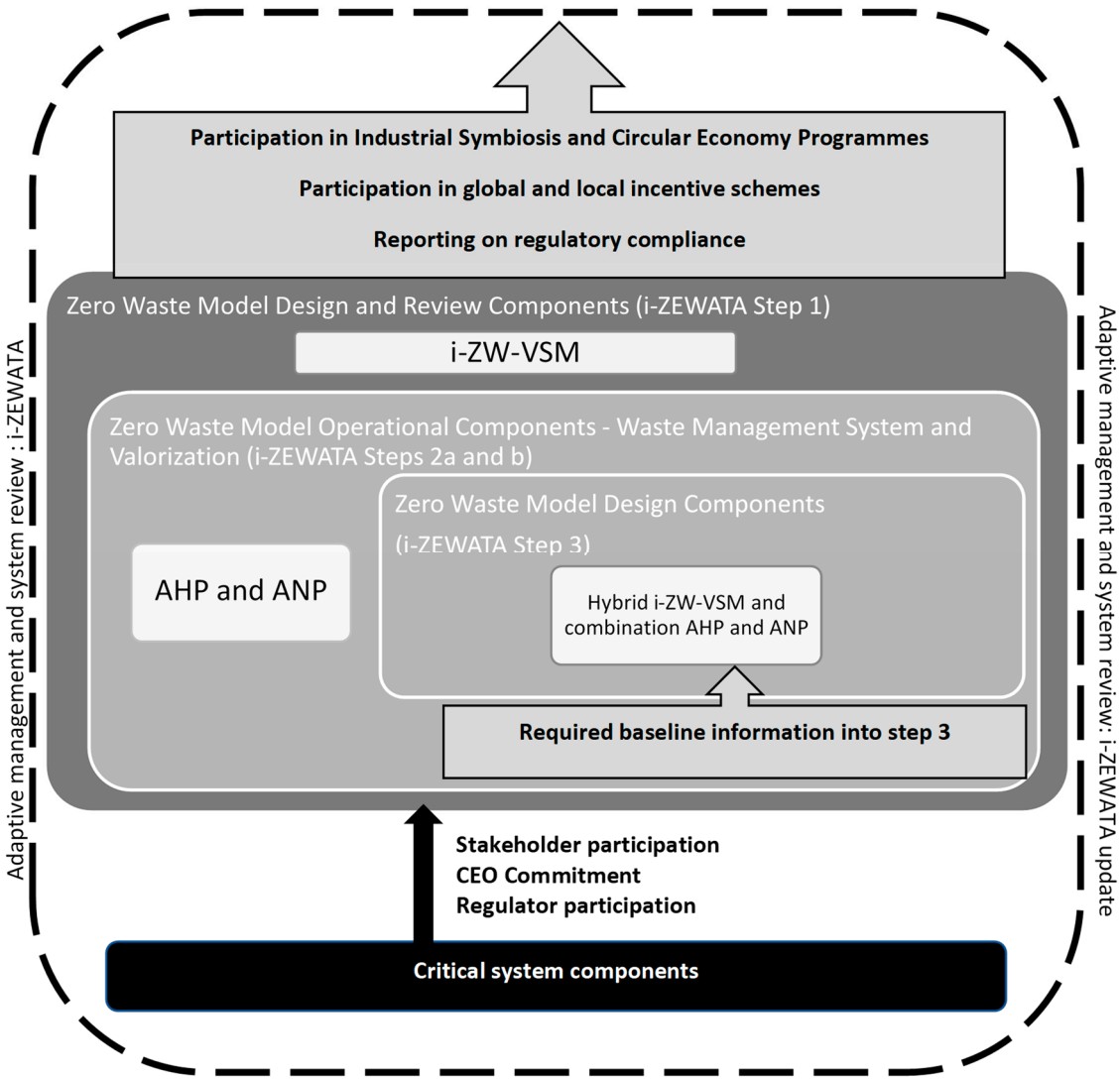

**Figure 1.** An overview of the proposed i-ZEWATA model.

### 2.1.1. Step 1: Determining the Status Quo of Industrial Waste Management

Step 1 of the i-ZEWATA model was concerned with applying the Value Stream Mapping methodology (VSM) [93] as an industrial zero-waste focused Value Stream Mapping (i-ZW-VSM) and consisted of two parts. The first part was concerned with data collection, and the second part was concerned with completing the VSM in three phases. Data collection consisted of auditing waste management facilities in terms of operational management and identifying key waste streams. Waste generation volumes were confirmed, and existing waste sampling data records were collected. Waste sampling was conducted on waste streams where limited data was available. Laboratory analyses were conducted on waste streams, and existing laboratory analysis records were used where available. The laboratory waste analyses were used to complete the risk profiling of the waste.

The VSM was completed in three key phases. Phase 1 included preparing and mapping waste fractions and waste generation and assessing waste streams. In Phase 2, a Horizontal performance analysis was conducted to determine the material or waste efficiencies for each segment. In Phase 3, a Vertical analysis was conducted on the iron and steel waste to determine the sub-process efficiencies to assist with the overall ZW analysis.

The specific data inputs required for Phase 1 analysis were waste generation volumes, information on waste streams, waste system costs and revenue, the nature of external services rendered, waste transportation modes, waste handling and movements, and

the waste treatment nature. A systems view was applied to waste material flow with a strong emphasis on ZW. The waste management system was divided into material and information flow [93] as specific sub-processes required in the VSM.

Assessing the level of waste efficiency achievement through waste reduction, waste avoidance, and limiting raw materials in the manufacturing process is critical in Phase 2. Phase 2 was also concerned with examining the waste management system and activities in detail and identifying all applicable waste activities. The Phase 2 analysis was separated into various main segments as representative waste types of process and general waste to understand the waste material flows and set key performance indicators (KPIs). Choosing the main segments were based on the case study's industrial operations that consequently generated different waste materials.

Calculating the waste efficiency (WE) can be expressed as a formula in the form of a valid approximation to material efficiency (ME) percentage. The waste efficiency (WE) percentage equals the weight of the product (PW) divided by the weight of waste material (WMW), in terms of the weight of the product (PW) divided by the sum of the waste (WMW) and product weight (PW)) Equation (1) that equals ME.

$$WE\ (\%) = PW/WMW \qquad ME\ (\%) \approx PW/(WMW + PW) \qquad (1)$$

Calculations conducted for the horizontal performance analysis included both general and process waste. The performance of each waste segment was monitored so that it will be possible to monitor the potential for improvements over time. The outcome of phase 2 further contributed to completing the current state VSM map and indicated the actual state of the waste management system.

Waste management sub-processes were used to analyze the critical subprocess performance measurements, namely cost efficiency, service efficiency, and overall efficiencies in Phase 3. The vertical analysis of the waste process and overall efficiency in each sub-process were then applied to general and process waste streams. To understand the average monthly performance and overall sub-process efficiencies, the vertical analysis results were presented as an averaged monthly performance measurement. Performance measures were conducted as an evaluation of the actual overall system effectiveness. For this purpose, vertical analysis of sub-processes had to be conducted. Following phase 3, the future state VSM was completed based on the ZW goals that had to be achieved. The actual state was aligned to the future state VSM.

The compilation of the VSM maps comprised the final step of the VSM. Using the analyzed waste data, an actual and future state map of waste management was compiled. Opportunities such as cost reduction, waste minimization improvements, waste reduction, environmental performance improvement, and time optimization were revealed by the actual and future state maps. Additionally, the maps provided essential input criteria and sub-criteria components to i-ZEWATA steps 2a and b.

### 2.1.2. Step 2a and b: Industrial Waste Management System and Valorization Components Prioritization

In Step 2a, industrial waste management system components were prioritized, and in Step 2b, waste valorization (including treatment) system components were prioritized. The outcome of step 2a was used as alternative priority input into step 3a.

#### Method: AHP

Steps 2a and b consisted of eight steps to complete the AHP to prioritize waste management system components. The AHP methodology was based on three fundamental principles: breaking down the problem, a pairwise comparison of the various alternatives, and a synthesis of the preferences. The outcome of step 2b was used as alternative priority input into step 3a.

Steps 2a and b can be summarized as follows:

1. Identifying criteria and sub-criteria for AHP input from the VSM outcome;

2. Weighing criteria;
3. Comparing criteria at the same level;
4. Completing the pairwise comparison matrix;
5. Checking consistencies;
6. Calculating local priorities;
7. Synthesizing the waste management system components; and
8. Constructing a decision matrix and interpreting the results.

The following AHP process, as adapted from [82] was followed to determine the waste management system and valorization components:

1. The decision-support problem, namely determining waste management system components, was broken down and structured as a hierarchy. The input information was obtained from the completed VSM method analysis. The lower levels were set as the intangible and or tangible criteria, and sub-criteria derived from the VSM contributed to achieving the primary goal. All defined criteria and weights were simulated in the form of a *matrix* $_n x_n$ (where $n$ is the number of the weights) comparing the criteria and weights with each other (Equations (2) and (3)).

$$
\left[ \widetilde{C}_w \right] = \begin{bmatrix} \dfrac{\widetilde{W}_1}{\widetilde{W}_1} & \cdots & \dfrac{\widetilde{W}_1}{\widetilde{W}_{n-1}} & \dfrac{\widetilde{W}_1}{\widetilde{W}_n} \\ \dfrac{\widetilde{W}_2}{\widetilde{W}_1} & \cdots & \dfrac{\widetilde{W}_2}{\widetilde{W}_{n-1}} & \dfrac{\widetilde{W}_2}{\widetilde{W}_n} \\ \cdots & \cdots & \cdots & \cdots \\ \dfrac{\widetilde{W}_n}{\widetilde{W}_i} & \cdots & \dfrac{\widetilde{W}_n}{\widetilde{W}_{n-1}} & \dfrac{\widetilde{W}_n}{\widetilde{W}_n} \end{bmatrix}. \tag{2}
$$

$$
\left[ \widetilde{C}_{\{\widetilde{A}_1, \ \widetilde{A}_{2,\dots,} \ \widetilde{A}_{1m} \ against \ \widetilde{C}_1\}} \right] = \begin{bmatrix} \dfrac{\widetilde{x}_{11}}{\widetilde{x}_{11}} & \cdots & \dfrac{\widetilde{x}_1}{\widetilde{x}_{m-1,1}} & \dfrac{\widetilde{x}_{11}}{\widetilde{x}_{m1}} \\ \dfrac{\widetilde{x}_{21}}{\widetilde{x}_{11}} & \cdots & \dfrac{\widetilde{x}_{21}}{\widetilde{x}_{m-1,1}} & \dfrac{\widetilde{x}_{21}}{\widetilde{x}_{11}} \\ \cdots & \cdots & \cdots & \cdots \\ \dfrac{\widetilde{x}_{m1}}{\widetilde{x}_{11}} & \cdots & \dfrac{\widetilde{x}_{m1}}{\widetilde{x}_{m-1,1}} & \dfrac{\widetilde{x}_{m1}}{\widetilde{x}_{m1}} \end{bmatrix} \tag{3}
$$

$\left[ \widetilde{C}_w \right]$ is the comparisonwise matrix of weights in which $\widetilde{w}_1, \widetilde{w}_2, \dots, \widetilde{w}_n$ were the defined weights of criteria 1 ($\widetilde{C}_1$), criteria 2 ($\widetilde{C}_2$), ..., criteria $n$ ($\widetilde{C}_n$), respectively. $\widetilde{C}_{\{\widetilde{A}_1, \ \widetilde{A}_{2,\dots,} \ \widetilde{A}_{1m} \ against \ \widetilde{C}_1\}}$ is the comparisonwise matrix of alternatives against criteria 1.

Against each criterion, the values of the alternatives were then compared according to Equation (3). All pairwise comparison matrices need to be reciprocal.

2. The weights of the criteria were obtained.

The sum of the values was calculated and normalized by the sum of all the rows' values. Within the matrix $\widetilde{C}$ the values were summed in each column. Their reciprocals (1/sum of the values of each column) were calculated. Next, by using (dividing) the sum of all reciprocals, the resulting reciprocal values were normalized. The values of each column were normalized by the sum of the values in the same column. Each row's average was calculated, which stood for the relative importance of alternatives against criteria 1. All the values in each row of the matrix $\widetilde{C}$ was multiplied, and those values' $n$th roots were normalized by the sum of the $n$th roots of those values. The normalized values indicated the relative importance of alternatives against the associated criteria in the comparisonwise matrix.

A value of 1 was produced when each of the four methods' relative importance values was summed.

$$
[\widetilde{A} \times \widetilde{w}] =
\begin{bmatrix}
\frac{\widetilde{W}_1}{\widetilde{W}_1} & \cdots & \frac{\widetilde{W}_1}{\widetilde{W}_{n-1}} & \frac{\widetilde{W}_1}{\widetilde{W}_n} \\
\frac{\widetilde{W}_2}{\widetilde{W}_1} & \cdots & \frac{\widetilde{W}_2}{\widetilde{W}_{n-1}} & \frac{\widetilde{W}_2}{\widetilde{W}_n} \\
\cdots & \cdots & \cdots & \cdots \\
\frac{\widetilde{W}_n}{\widetilde{W}_i} & \cdots & \frac{\widetilde{W}_n}{\widetilde{W}_{n-1}} & \frac{\widetilde{W}_n}{\widetilde{W}_n}
\end{bmatrix}
\begin{bmatrix}
\widetilde{w}_1 \\
\widetilde{w}_2 \\
\cdots \\
\cdots \\
\cdots \\
\widetilde{w}_{n-1} \\
\widetilde{w}_n
\end{bmatrix}
= \lambda_{max}
\begin{bmatrix}
\widetilde{w}_1 \\
\widetilde{w}_2 \\
\cdots \\
\cdots \\
\cdots \\
\widetilde{w}_{n-1} \\
\widetilde{w}_n
\end{bmatrix}
\tag{4}
$$

$$
= \lambda_{max} \widetilde{W}
$$

$\lambda_{max}$ is comparison matrix $C_a$'s largest eigen value. Eigen vector w was used to prioritize or weigh the alternatives (Equation (3)).

Within the comparison matrix, $\widetilde{C}_w = \widetilde{a}_{ij}$, $\widetilde{a}_{ij} = \frac{\widetilde{w}_i}{\widetilde{w}_j}$ for $i, j = 1, 2, \ldots, n$, and $\widetilde{a}_{ij} = 1/\widetilde{a}_{ij}$, indicated the reciprocal matrix status of comparison matrix A. Further, if the condition in Equation (5) is met, comparison matrix A will be consistent Equation (4)

$$
\widetilde{a}_{jk} = \widetilde{a}_{ik}/\widetilde{a}_{ij}
\tag{5}
$$

where $i$, $j$, and $k = 1, \ldots, n$.

3. Calculating the consistency index (CI).

The Consistency Index (CI) is a measure of inconsistency for a pairwise comparison, and CI was used to determine the consistency ratio. CI was calculated as shown in Equation (6).

$$
\text{CI} = (\lambda_{\max} - n)/(n - 1)
\tag{6}
$$

where $\lambda_{\max}$ is the maximum eigenvalue of the comparisonwise matrix of $\widetilde{C}_a$ and $n$ is the dimension of the comparisonwise matrix of $\widetilde{A}$.

Based on CI, the consistency ratio was calculated as in Equation (7):

$$
CR = \frac{\text{CI}}{RI}
\tag{7}
$$

4. Finding the weight vector for each pairwise comparison matrix

Normalized eigenvectors served as priority weights for alternatives in the comparison matrix as long as the consistency ratio was less than 0.10. The fourth method provided values that were significantly near to eigenvector values; therefore, the $n$th roots of the multiplicative values in each row of the comparison matrix after normalization indicated the considered alternatives' priority weights in each comparisonwise matrix of alternatives against each criterion [94].

5. By using the priorities of the bottom-level criteria and alternatives, the decision matrix was developed.
6. The AHP method (weighted sum model) was used to aggregate the alternative priorities and criteria' priorities.

### 2.1.3. Step 3a: Development of a ZW Multi-Criteria Decision-Support Model

Method: AHP (as combined AHP and ANP)

In step 3a, the first method of the combined approach was an AHP method and consisted of the following key steps:

1. Developing alternative priorities for each criterion Step 2a and 2b;
2. Comparing alternatives for each bottom level criterion;
3. Completing the pairwise comparison matrix;
4. Checking the consistency;

5. Calculating the alternative priorities for each bottom-level criterion; and
6. Compiling the decision matrix.

The outcome of step 3a was used as a decision matrix input into the ANP in step 3b.

### 2.1.4. Step 3b: Development of a ZW Multi-Criteria Decision-Support Model

Method: ANP (as combined AHP and ANP)

In step 3b, the first method of the combined approach was an ANP method and consisted of the following key steps:

1. Identifying elements and clusters from the decision matrix in AHP;
2. Identifying relationships;
3. Identifying elements influencing the pairwise comparisons;
4. Identifying clusters influencing pairwise comparisons;
5. Compiling the relationship matrix;
6. Compiling the unweighted supermatrix;
7. Developing the cluster supermatrix;
8. Developing the weighted supermatrix;
9. Normalizing the supermatrix;
10. Limiting the supermatrix and priorities;
11. Interpreting and developing the ZW model.

The combined AHP (step 3a) and ANP (step 3b) methods were applied by using the software "Super Decisions" (Version 3.2., Creative Decisions Foundations, Pittsburgh, PA, USA). The criterion weighting method applied as the AHP method combined equal, subjective, and objective weighting methods. Seven steps were applied in developing the ZW multi-criteria decision-support model.

Step 1: The combined outcome of steps 2a and b was broken down into a hierarchy of goals, criteria, and alternatives. Two separate AHP processes of steps 2a and b were used as input priorities for alternative criteria (Equation (8)) as adapted from [72].

$$
\begin{aligned}
\left[ \widetilde{A} \times \widetilde{w} \right] = \begin{bmatrix} \frac{\widetilde{W}_1}{\widetilde{W}_1} & \cdots & \frac{\widetilde{W}_1}{\widetilde{W}_{n-1}} & \frac{\widetilde{W}_1}{\widetilde{W}_n} \\ \frac{\widetilde{W}_2}{\widetilde{W}_1} & \cdots & \frac{\widetilde{W}_2}{\widetilde{W}_{n-1}} & \frac{\widetilde{W}_2}{\widetilde{W}_n} \\ \cdots & \cdots & \cdots & \cdots \\ \frac{\widetilde{W}_n}{\widetilde{W}_i} & \cdots & \frac{\widetilde{W}_n}{\widetilde{W}_{n-1}} & \frac{\widetilde{W}_n}{\widetilde{W}_n} \end{bmatrix} \begin{bmatrix} \widetilde{w}_1 \\ \widetilde{w}_2 \\ \cdots \\ \cdots \\ \cdots \\ \widetilde{w}_{n-1} \\ \widetilde{w}_n \end{bmatrix} = \lambda_{max} \begin{bmatrix} \widetilde{w}_1 \\ \widetilde{w}_2 \\ \cdots \\ \cdots \\ \cdots \\ \widetilde{w}_{n-1} \\ \widetilde{w}_n \end{bmatrix} \\
= \lambda_{max} \widetilde{W} \\
\left[ \widetilde{A} \times \widetilde{w} \right] = \begin{bmatrix} \frac{\widetilde{W}_1}{\widetilde{W}_1} & \cdots & \frac{\widetilde{W}_1}{\widetilde{W}_{n-1}} & \frac{\widetilde{W}_1}{\widetilde{W}_n} \\ \frac{\widetilde{W}_2}{\widetilde{W}_1} & \cdots & \frac{\widetilde{W}_2}{\widetilde{W}_{n-1}} & \frac{\widetilde{W}_2}{\widetilde{W}_n} \\ \cdots & \cdots & \cdots & \cdots \\ \frac{\widetilde{W}_n}{\widetilde{W}_i} & \cdots & \frac{\widetilde{W}_n}{\widetilde{W}_{n-1}} & \frac{\widetilde{W}_n}{\widetilde{W}_n} \end{bmatrix} \begin{bmatrix} \widetilde{w}_1 \\ \widetilde{w}_2 \\ \cdots \\ \cdots \\ \cdots \\ \widetilde{w}_{n-1} \\ \widetilde{w}_n \end{bmatrix} = \lambda_{max} \begin{bmatrix} \widetilde{w}_1 \\ \widetilde{w}_2 \\ \cdots \\ \cdots \\ \cdots \\ \widetilde{w}_{n-1} \\ \widetilde{w}_n \end{bmatrix} \\
= \lambda_{max} \widetilde{W}
\end{aligned}
\tag{8}
$$

Using the bottom-level criteria and alternatives priorities, the decision matrix was developed for input into the ANP. The AHP method (weighted sum model) was applied to aggregate the alternative priorities and criteria priorities.

Following the decision model's construction and establishing relations between the elements, the pairwise comparisons between elements were determined. This evaluation took place at two levels: (1) that of clusters, and (2) that of nodes using the absolute scale [95], which translated verbal reviews and numerical ratings. The assigned ratings were placed in a pairwise comparison.

The elements' relationships were identifiedand the (N × N) Elements' Relationships matrix (Equation (9)),

$$= [r_{i,j}] = \left[ r_{i,j}^{c_a,c_b} \right] \text{ obtained.} \tag{9}$$

- $r_{i,j}^{c_a,\,c_b} \in \{0,1\}$ where $C_a, C_b = 1, \dots, G$ and $I, j = 1, \dots, N$:

- $r_{i,j}^{C_a,C_b} = 0$ indicates that the element $x_i^{C_a}$ has no influence on the element $x_j^{C_b}$, and in the graphical model, there is not an edge between $x_i^{c_a}$ and $x_j^{C_b}$.

- $r_{i,j}^{C_a,C_b} = 1$ indicates that the element $x_i^{c_a}$ has some influence on the element $x_j^{C_b}$, and in the graphical model, there is an arc from $x_i^{c_a}$ to $x_j^{C_b}$.

Step 2: Priorities were derived for the criteria.

The criteria' importance was compared pairwise concerning the ZW model development's goal and to derive their weights. The consistency was checked. A review was undertaken to ensure a reasonable level of consistency in terms of proportionality and transitivity.

The normalized eigenvectors served as priority weights for alternatives in the comparison matrix, where the consistency ratio was less than 0.10. The applied method in the AHP provided values that were significantly near to eigenvector values. After normalization, the *n*th roots of the multiplicative values in each row of the comparison matrix indicated the considered alternatives' priority weights in each comparisonwise matrix of alternatives against each criterion.

Step 3: Priorities were derived for the alternatives.

Priorities were derived for each alternative concerning each criterion (Equation (10)). A similar process was followed as in step 2, where alternatives were pairwise compared concerning each criterion.

The (G × G) Clusters' Relationship matrix, $\hat{R} = \left[ \hat{r}_{c_a,\,c_b} \right]$ was obtained.

$$\hat{r}_{c_a,c_b} = \{0,1\} \text{ where } c_a,c_b = 1, \dots, G : r_{i,j}^{C_a,C_b} = 0$$

- $\hat{r}_{c_a,\,c_b} = 0$ indicate that any element of cluster $c_a$ has influence on any element of cluster $c_b$.

$$\hat{r}_{c_a,\,c_b} = 0 \rightarrow \forall i, j \qquad i,j = 1, \dots, N : r_{i,j}^{c_a,\,c_b} = 0$$

- $\hat{r}_{c_a,c_b} = 1$ indicate that some element of cluster $c_a$ has influence on some (at least one) elements of cluster $c_b$.

$$\hat{r}_{c_a,c_b} = 1 \rightarrow \exists i, j \qquad i,j = 1, \dots, N : r_{i,j}^{c_a,\,c_b} = 1 \tag{10}$$

The separate AHP pairwise matrices (Equation (8)) were used to compare the influence of the elements of the combined alternatives belonging to each of the clusters on any element. A priority vector was derived, and the (N × N) Unweighted Supermatrix was obtained, $U = \left[ u_{i,j}^{c_a,c_b} \right]$, with $u_{i,j}^{c_a,c_b} \in [0,1]$, $c_a,c_b = 1, \dots, G$ and $i, j = 1, \dots, N$, where $u_{i,j}^{c_a,c_b}$ is the influence of element I, which belongs to cluster $c_a$, on element $j$, which belong to cluster $c_b$ (Equation (11))

$$u_{i,j}^{c_a,c_b} = 0 \leftrightarrow r_{i,j}^{C_a,C_b} = 0 \tag{11}$$

- $u_{i,j}^{c_a,c_b} = 1$ indicated that the element *I*, which belongs to cluster $c_a$ is the unique element of cluster $c_a$ which influences element $j$, which belongs to cluster $c_b$ (Equation (12))

$$u_{i,j}^{c_a,c_b} = 1 \implies \forall k \neq i, \qquad k = 1, \dots, N : x_k \in x^{c_a} \rightarrow r_{i,j}^{C_a,C_b} = 0 \tag{12}$$

Given cluster, $c_a$, and an element j that belongs to cluster $c_b$, $x_j^{c_b}$, the sum of the unweighted values of the elements which belong to $c_a$, influence $x_j$ is 1. Where elements of $c_a$ influences $x_j$ the sum is 0 (Equation (13)).

Given $c_a$ $x_j^{c_b}$

$$\sum_{\substack{k=1 \\ k:x_k \in x^{c_a}}}^{N} \left( u_{k,j}^{C_a,C_b} \right) \in \{0,1\} \tag{13}$$

Columns, $\sum_{i=1}^{N} (u_{i,j})$, indicated how many clusters influence the column element. The components and elements of the network and their relationships were identified.

Pairwise comparisons on the clusters were conducted, obtaining $\hat{U} = \left[ \hat{U}_{c_a,c_b} \right]$ the $(G \times G)$ Cluster Weights matrix with $\hat{U}_{c_a,c_b} \in [0,1]$, $c_a, c_b = 1, \dots, G$, where $\hat{U}_{c_a,c_b}$ is the influence of cluster $c_a$ on cluster $c_b$ (Equations (14)–(16))

- $\hat{U}_{c_a,c_b} = 0$, indicated that any element of cluster $c_a$ influences any element of cluster $c_b$.

$$\sum_{c_a=1}^{G} (U_{c_a,c_b}) = 1 \tag{14}$$

$W = W_{i,j}^{C_a,C_b}$ was calculated, the $(N \times N)$ Weighted Supermatrix, with $W_{i,j}^{C_a,C_b} \in [0,1]$, $c_a, c_b = 1, \dots, G$ and $I, j = 1, \dots, N$, where $W_{i,j}^{C_a,C_b} = u_{i,j}^{C_a,C_b} \times \hat{U}_{c_a,c_b}$ (15)

- $W_{i,j}^{C_a,C_b}$ is the weighted influence of element $I$, which belongs to cluster $c_a$, on element $j$, which belongs to $c_b$.

$$\sum_{i=1}^{N} \left( W_{i,j}^{C_a,C_b} \right) \in [0,1] \tag{16}$$

As with the ANP, a supermatrix represents the relationship between the relative assigned weights of a network model. In the network model, all the priority vectors extracted from individual pairwise comparisons matrices through an AHP were used to complete the ANP analysis.

Three different supermatrices were extracted during the development of the ANP methodology in step 4, namely:

- The un-weighted supermatrix or initial supermatrix contained all the eigenvectors that were derived from the pairwise comparison matrices of the model;
- The weighted stochastic supermatrix was obtained by multiplying the values in un-weighted supermatrix by each cluster's weight. The priority level assigned to each cluster was considered;
- The limit weighted final supermatrix was obtained where the supermatrix was raised to a limiting power to obtain and converge a stable set of weights that represented the final priority vector.

As a final check, data were checked and adjusted according to the consistency required.

Step 4: Synthesizing the model

It was necessary to synthesize the outcome of alternative priorities and structures to obtain their overall synthesis. The final step in the process was to converge the weighted supermatrix to obtain a long-term stable set of weights. In Equation (17), the supermatrix was raised to a limit lower to obtain a matrix where all the columns each provided the global priority vector and were equal:

$$\lim_{k \to \infty} W^k \tag{17}$$

All alternative priorities obtained were then combined as a weighted sum, considering each criterion's weight to establish the alternatives' overall priorities. The alternatives with the highest overall priority constituted the best choice.

Step 5: Performing sensitivity analysis

A study was completed on criteria weight changes to understand the rationale behind obtaining the results.

Step 6: Final decision making

A decision was made based on the alternatives and criteria accompanying the best alternative in the development of a ZW model for the iron and steel facility.

Step 7: Model construction

The final step in developing a ZW multi-criteria decision-support model was to construct step six's outcome into a framework model guiding ZW design and implementation.

## 3. Case Study Results and Discussion

This section demonstrates the application of the developed i-ZEWATA model methodology for developing a ZW multi-criteria decision-support model. A vertically integrated South African iron and steel manufacturing facility, founded in 1957, comprised the case study. The iron and steel facility produces around 1 million tons of steel blocks annually. The case study was selected because of the availability of actual recorded data and the presence of legacy waste facilities and -activities. The facility's industrial waste management was further aggravated by local sustainability challenges, complex and isolated legacy waste management operations and systems due to the complex nature of the waste streams (and the facility) that further caused misinformed decision-making.

### 3.1. Step 1: Baseline Assessment and Industrial ZW—Value Stream Waste Flow Mapping

In step 1, the baseline environment was investigated, and a site-specific assessment was conducted. A VSM mapping comprised step 1 and included three phases. The first phase included waste generation and fraction mapping. In the second phase, a horizontal performance analysis of materials efficiency per main waste segment was conducted. In the third phase, a vertical analysis was conducted of the waste management process efficiency in each sub-process. To provide an overview of performance parameters associated with waste generation quantities, waste costs, and waste flows, an actual and future state VSM was compiled.

The results from the completed VSM are illustrated in Tables 1 and 2 and can be used as input to compile actual and future state VSM maps (Figure 2). The actual VSM map (Figure 2) can be used as a process flow diagram of the iron and steel waste and, hence, be used as a baseline diagram.

The results indicate that 0.4 tons of general waste are recycled as compared to 0.22 tons of process waste per ton of waste generated (recycling rate of 22%). There are also significant differences in overall efficiencies relating to the external treatment of waste when comparing the overall efficiency of general waste USD 0.28 and process waste of USD 1.36 per ton of crude steel produced (Tables 1 and 2). The differences in overall efficiencies are related to the nature of on and off-site waste treatment and external disposal costs (as is the case for general waste). However, in the case of process waste the overall efficiency cost is a helpful indicator of the limited internalization of waste that is taking place costs due to on-site waste treatment and disposal with a limited focus on CIWM (Table 2).

An initial baseline ZW waste annual reduction percentage of 5% is proposed based on the VSM conducted in South Africa that considers the local environmental, economic, technology, regulatory, and social components. The 5% ZW waste reduction percentage is recommended to be implemented across the waste management system as a 5% rule for the baseline upon which subsequent improvements can be made. The 5% is realistically achievable and will not merely be tactics of diversion.

**Table 1.** General waste VSM results in the case study averaged over six years (adapted from [96]).

| Subprocess Performance Actual Measurements (Monthly) | Containers (Cn) | | Handling (Internal) | | Transportation (External) | | Treatment (External) | |
|---|---|---|---|---|---|---|---|---|
| Service Efficiency | # (Cn)/W (waste in bins) | 60 Cn/110 tons | Person-h/W | 880/110 tons | # (trucks)/W (waste transported) | 5/110 | W (recycled)/W (sum)(sum) | 349 tons/872 tons |
| | 0.55 Cn available/ton of generated general waste | | 8 person-hours required to manage one ton of generated general waste | | 0.05 trucks available/ton of general waste generated | | 0.4 tons are recycled for every ton of general waste generated | |
| Cost Efficiency (Unit of Cost expressed in USD) | Cn /W (waste in bins) | 11,905/110 tons | C (Person)/W | 14,881/110 tons | C (transport+ disposal)/W (waste transported) | 15,739/110 tons | C (treatment − disposal & transport)/W (sum) | 15,739/110 tons |
| | Costs USD 108 to maintain Cn/ton of general waste | | USD 135 as labor costs to manage one ton of general waste generated | | To transport and treat one ton of general waste, the cost is USD 143 | | | |
| Overall Effectiveness (Unit of Cost expressed in USD) | Cn /P | 11,905/57,219 tons | C (Person − h)/P | 14,881/57,219 | C (trucks)/W (waste transported) | 15,739/110 tons | C (treatment)/P | 15,739/57,219 |
| | Costs USD 0.21 to maintain Cn/ton of produced crude steel | | The labor costs are USD 0.26 per ton of crude steel produced | | USD 143 as treatment and transport costs/ton of generated general waste | | Costs of USD 0.28/ton of crude steel to transport and treat one ton of general waste | |

**Table 2.** Process waste VSM results in the case study averaged over six years (adapted from [96]).

| Subprocess Performance Actual Measurements Per Month | Disposal Facilities (On-Site) | | Handling (Internal) | | Treatment (Internal) | |
|---|---|---|---|---|---|---|
| Service Efficiency | # (disposal facilities)/W (waste generated) | 46 facilities /125,839 tons | Person-h/W | 720/125,839 | W (recycled)/W (sum) (sum) | 27,439 tons/125,839 tons |
| | 0.0004 on-site facilities available/per ton of process waste generated | | For each ton of waste generated, 0.006 person-hours are available to deal with such waste | | For each one ton of waste generated, 0.22 tons of waste is recycled or reused | |
| Cost Efficiency (Unit of Cost expressed in USD) | C (disposal facilities)/W (waste generated) | 77,958/125,839 tons | C (person)/W | 14,881/125,839 tons | C (treatment − disposal & transport)/W (sum) | 77,958/125,839 tons |
| | Costs USD 0.62/ton to manage process waste | | Costs USD 0.12 (as labor costs)/ton of process waste | | Costs USD 0.62/ton to manage process waste | |
| Overall Effectiveness (Unit of Cost expressed in USD) | C (disposal facilities)/P | 77,958/57,219 tons | C (person − h)/P | 14,881/57,219 | C (treatment)/P | 77,958/57,219 |
| | Costs USD 1.36/ton of crude steel produced to manage and dispose of waste | | Costs USD 0.26 (as labor costs)/ton of crude steel produced | | Costs USD 1.36/ton of crude steel produced to manage, treat, and dispose of waste | |

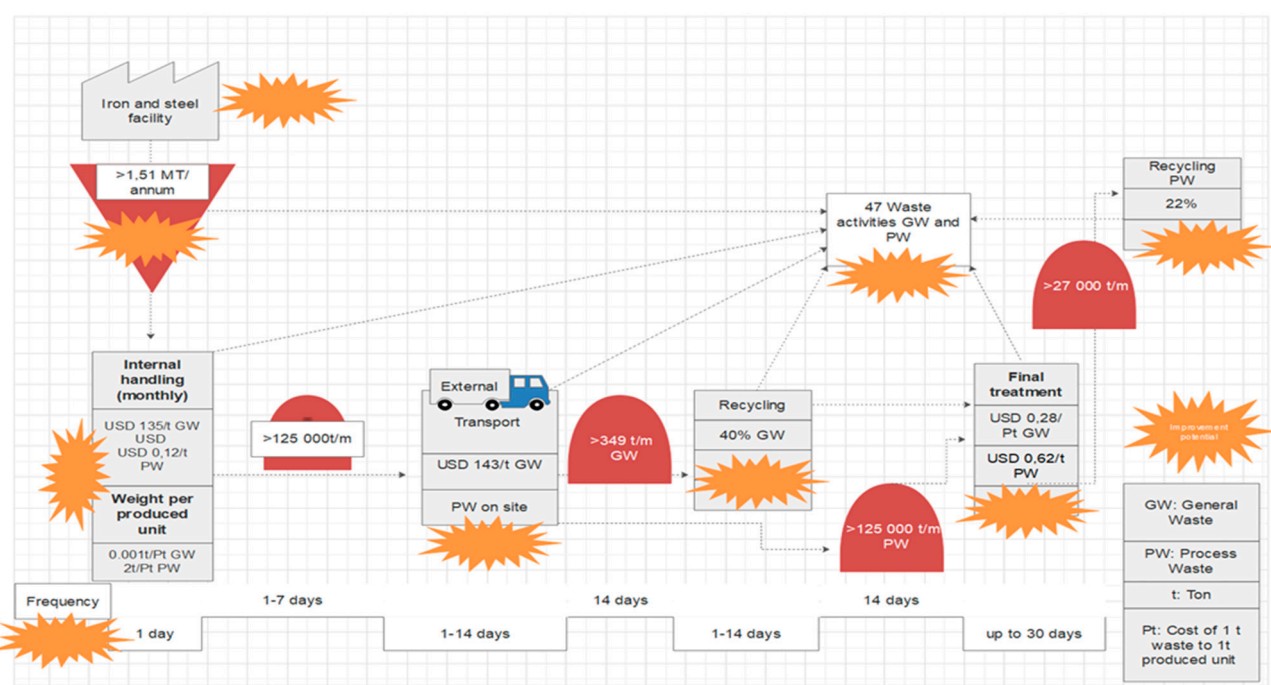

**Figure 2.** Actual initial state VSM of the iron and steel facility over a one-year period [96].

*3.2. Step 2a: Iron and Steel Waste Management System Component Prioritization (Database iZEWATA 0203)*

In step 2a, an integrated process was followed, where the baseline information generated in step 1 was used as input criteria for step 2a. The waste management system and treatment components of the case study were assessed using AHP methodology that included criteria identification and analysis, criteria weighting and priority identification. The pairwise comparison matrix for the main clusters applicable to prioritize and develop a waste management system is illustrated in Table 3.

**Table 3.** Main cluster weights applicable to waste management system component prioritization.

| Criteria | Weights | Priority |
|---|---|---|
| Company Culture | 0.20503 | 1 |
| Cost of Iron and Steel Waste | 0.08832 | 7 |
| Environmental Externalities | 0.09703 | 6 |
| Plant Divisions | 0.03755 | 8 |
| Regulatory | 0.17846 | 2 |
| Iron and Steel Waste Data System | 0.15698 | 3 |
| Iron and Steel Waste Infrastructure | 0.1086 | 5 |
| Iron and Steel Waste Streams | 0.12803 | 4 |

The pairwise comparison indicates that company culture is the main priority in developing a waste management system. Regulatory aspects such as site and general legal requirements rates as priority number 2 with the waste data system rating as priority number 3. The results indicate that a waste management system needs to be supported by the highest level of authority, such as the Chief Executive Officer (CEO), to implement successfully. Noteworthy for prioritization include compliance with regulatory requirements

and developing and maintaining a credible waste data system consisting of collection, monitoring, recording, analysis, storage, and reporting protocols. Another priority cluster identified is identifying waste streams that influence treatment and disposal options and regulatory requirements compliance, reworking options, waste externalization, waste exchange, and waste internalization.

### 3.3. Step 2b: Iron and Steel Waste Valorization System Component Prioritization (Database iZEWATA 0203)

In step 2b, information gathered in steps 1 and 2a was used as input criteria to complete step 2b through a combined AHP and ANP methodological approach that included criteria identification and analysis, criteria weighting, priority identification, and sensitivity analysis. The model can then be monitored and evaluated by recording various Industrial Zero Waste (i-ZW) indicators. The pairwise comparison matrix for the main clusters applicable to prioritize and develop a waste treatment system is illustrated in Table 4.

**Table 4.** Main cluster weights applicable to the prioritization of a waste valorization system.

| Criteria | Weights | *Priority* |
|---|---|---|
| Contaminant Management | 0.17227 | 2 |
| Management Practices | 0.26178 | 1 |
| Disposal and Treatment Methods | 0.05726 | 7 |
| Regulatory Framework | 0.13370 | 5 |
| Site Remediation | 0.07941 | 6 |
| Sustainability and Circular Economy Practices | 0.14013 | 4 |
| Waste Treatment Cost | 0.15545 | 3 |

Results from the AHP pairwise comparison indicate that valorization management practices, including the financial viability thereof, comprise the top priority in identifying waste treatment system requirements, options, and opportunities (including industrial ecology) available for processing the various components of the waste stream. Opportunities include alternatives to landfilling, to minimize the waste stream and provide options to divert waste from landfills. By developing and implementing management practices, waste streams can be characterized and better understood to promote efficient contaminant management. Also, an improved understanding of the economic implications will be an outcome as associated with the different types of waste generated on-site that need to be treated effectively. Contaminant management is also a top priority (number 2). It includes risk classification, community impact, understanding the fate and transport of contaminants, and the management of air, water, and soil discharges. Site remediation achieved a priority score of 6 due to financial constraints and the limited amount of capital available to expedite extensive site remediation.

Another top priority cluster criterion is waste treatment costs. Once waste streams are characterized, and a waste valorization system developed, contamination can be managed and mitigated, and consequently, costs associated with treating waste quantified and valorization optimized.

### 3.4. Step 3: Iron and Steel ZW Management Model Development (Database iZEWATA 0203)

The alternative analysis (Database I-ZEWATA 0203) indicates that for Step 2, the alternative ZW achieved a score of 42.3%. In contrast, the Best Practice approach scored 29.1%, the Compliance approach scored 22.3%, and finally, the Business-as-Usual approach scored 6.1%. Therefore, following the hybrid AHP and ANP analysis, the ZW approach is the most suitable for iron and steel waste management systems. The alternative analysis

results indicate that for Step 3 (Database I-ZEWATA 0203), the alternative ZW has a score of 47.4%. In comparison, the Best Practice approach scored 30.2%, the Compliance approach scored 16.8%, and finally, the Business-as-Usual approach scored 5.4%. Therefore, by applying the hybrid i-ZEWATA model methodology, the ZW approach was found to be the most suitable industrial waste management alternative for iron and steel waste management systems in developing countries in Africa.

### 4. A ZW Multi-Criteria Decision-Making Model for the Iron and Steel Industry

The ZW model is illustrated in Figure 3. The ZW model is suitable for implementing in iron and steel industries that have not implemented any ZW initiatives or have implemented ZW initiatives. The ZW model's initial implementation is suitable for industries that have not yet implemented the ZW model. The initial implementation process consists of seven steps (Table S1). Each step has been divided into the ZW model implementation framework components, input followed by the suggested methods, components, tools, or data for the required input and the output.

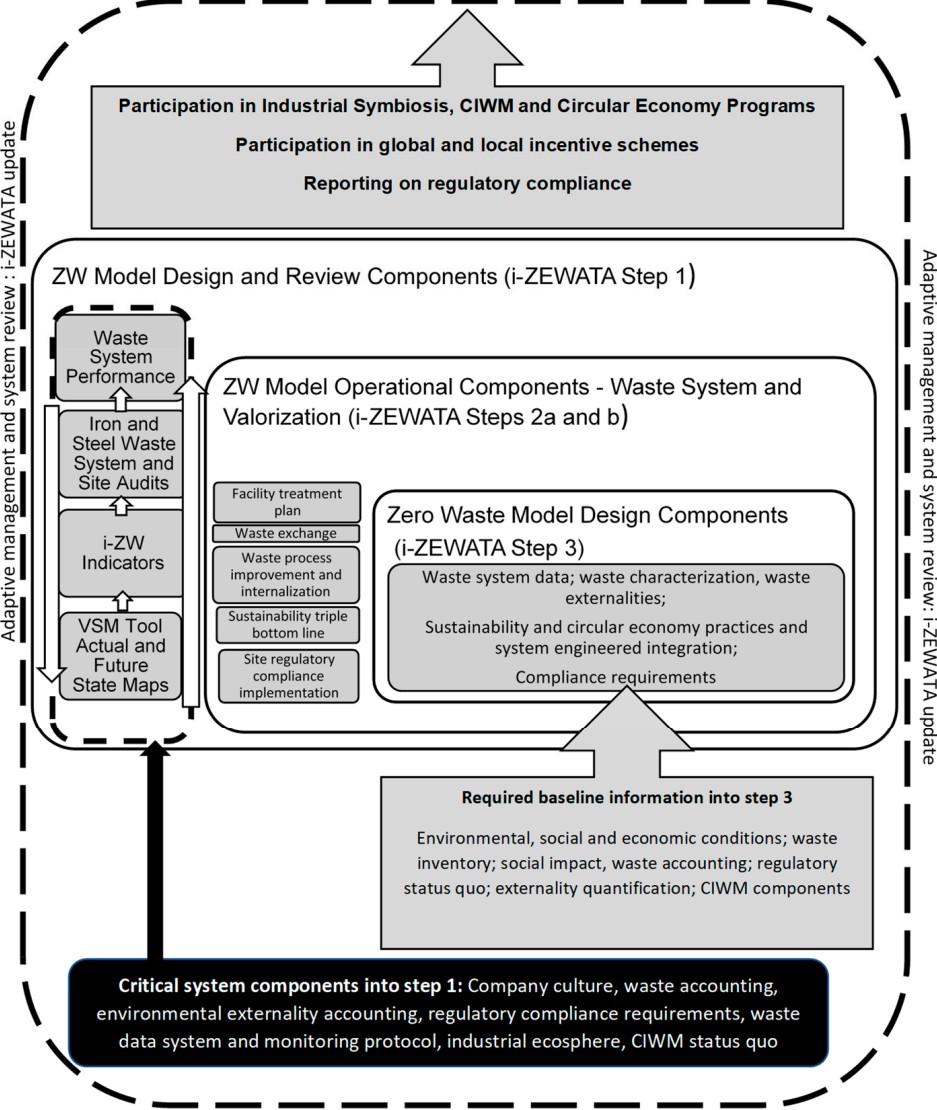

**Figure 3.** The ZW model for the iron and steel industry.

A ZW Model needs to adapt to changing technology, local economic, social, and environmental conditions, changing regulatory requirements, waste valorization opportunities and challenges, and CIWM approaches and opportunities. Stakeholder participation

should be integrated over the ZW model process's complete flow, together with participation from regulatory agencies. The ZW model should be adopted by the highest level of authority in the company, the CEO.

The ZW system engineered model's goal functions as a CIWM SCP tool that aims to promote a thorough analysis of industrial waste management to assist in the decision-support process of developing and implementing CIWM systems in the circular economy. In developing countries, the ZW model can address uncertainty in complex industrial waste systems in the iron and steel industry while promoting the United Nation's Sustainable Development Goal 12 as SCP. The ZW model couples a system engineering model with system assessment tools in a hybrid configuration. The model also addresses the variable of waste composition and becomes a precursor to developing an optimal, adaptive and resilient CIWM system for the iron and steel industry in developing countries. The ZW model can also be applied in the design process and assessment of CIWM from a systems perspective. The outcome of implementing the ZW model through the i-ZEWATA methodology can deliver on creating waste exchange opportunities in industrial symbiosis and waste internalization specifically applicable to legacy manufacturing facilities where waste avoidance and minimization are not always feasible due to aging manufacturing technology and facility layouts. Additionally, the ZW model, through promoting decision-support, can promote full CIWM accounting by connecting horizontal and vertical subprocesses to determine the connections between waste generation and waste treatment and therefore not only prepare industrial facilities to participate in industrial symbiosis programs but also to measure the readiness of a particular facility to participate in such programs.

Guidance on initial implementation of the ZW model and ZW model performance assessment and review after implementation is indicated in the Supplementary Material, Table S1. The ZW model performance assessment and review can be conducted annually to review the ZW model regarding social, technological, legal, economic, environmental applicability, circular economy program implementation, practicality, and progress aspects. The ZW model performance assessment and review can only be done after the ZW model's initial implementation or after the implementation of existing ZW initiatives.

The ZW model implementation and performance assessment and review consist of 7 steps (Supplementary Material, Table S1). The suggested quantitative monitoring components, as i-ZW indicators, that comprise step 6 of the ZW model implementation and performance review assessment in Table 5 are essential to prepare iron and steel facilities to participate in waste exchange and industrial symbiosis programs by (a) mapping, (b) interpreting, and (c) understanding site conditions, (d) identifying potential waste streams, (e) monitoring performance of waste systems and (f) unlocking value in waste. The i-ZW indicators' purpose is to monitor ZW system performance according to predefined indicators relating to the environment and sustainability, cost, management, culture, safety, and compliance.

ZW model development, implementation and review requires a blueprint or master plan to be made available for facility complex management and valorization. This plan needs to be reviewed annually or when significant system and treatment changes are required. The complex management plan includes both system and valorization management components, and the proposed components are indicated in the Supplementary Material, Table S1.

**Table 5.** i-ZW indicators for ZW model implementation monitoring and performance review assessment.

| Environmental Indicators | Waste Cost | Waste Management | Company Culture | Operator Safety | Compliance |
|---|---|---|---|---|---|
| Total water consumption (ton/unit) | General waste management (USD/ton) | Waste diverted from landfill (ton) | Waste training (*n* staff/total staff) | | |
| The ratio of use of waste material vs. virgin and non-waste as input materials (% or ton) | Process waste management (USD/ton) | Waste internalized (ton) | Community and internal complaints (number) | | |
| Waste internalized (ton) | Labour cost (USD/unit) | Waste externalized (ton) | | | |
| Waste discharged (ton) | Waste treatment cost (USD/ton) | General waste generation (ton) | | | |
| Green House Gas emissions (ton $CO_2$ eq/ton) | Waste exchange income (USD/ton and total USD income received) | Process waste generation (ton) | CEO waste system interaction or initiatives (number/total CIWM initiatives on facility-level) | Injury rate (injuries/unit) | Compliance with site regulations (%) |
| Dust levels (mg/m$^3$) | Waste internalization and externalization costs (USD/ton) | Waste recycled (ton) | | | |
| Soil remediation (m$^3$) | Legal compliance and penalty costs (USD) | Waste Facility Airspace remaining (m$^3$) | | | |
| | | Total waste disposed to landfill (on and off-site intons) | | | |
| Bioremediation (ton or m$^3$) | Waste discharge charges (USD/m$^3$) | Total amount of waste valorized (ton) | | | |

## 5. Conclusions

Managing iron and steel waste systems differ significantly from municipal waste systems. Implementing a ZW model as a ZW multi-criteria decision-support model in developing countries differs significantly from implementing a ZW model in developed countries. The social needs, economic conditions, required company performance, environmental conditions, and remediation requirements in developing countries differ from those in a developed country.

The ZW model encourages participation from all stakeholders in the manufacturing operation's immediate area, promoting participation, inclusivity, and accountability. Company culture is of cardinal importance, as it influences the implementation efficiency and adoption of the ZW model. Waste and environmental accounting form a crucial part of the ZW model to quantify environmental savings and the net environmental burden versus the net environmental benefit. Dealing with legacy waste disposal facilities and the subsequent ongoing contamination resulting from unlined facilities remains a challenge at many iron and steel manufacturing sites in developing countries. However, with the ZW model's implementation, a CIWM system can be developed where a relationship is established between the circular economy and SCP tools to move away from the business-as-usual and compliance-based industrial waste management approach and to embrace circular economy concepts. Embracing circular economy concepts can promote achieving the maximum economic profits and society benefits at the expense of reduced natural resources consumption and minimum environmental impacts.

Implementing the i-ZEWATA methodology as part of the ZW model can provide a decision-support tool to assist sustainability, waste, and environmental managers with specific functionalities to understand, visualize, map, interpret and communicate industrial waste flows and develop facility appropriate CIWM plans and programs. Implementing the i-ZEWATA methodology as part of the ZW model can promote a substantial change in environmental behavior by encouraging the adoption of circular practices in CIWM. Further, by identifying the fundamentals of industrial waste management challenges, greenwashing can be counteracted in the iron and steel industry. The model integrates

socio-economic and environmental factors and selects, prioritizes, and provides for comparing waste management alternatives. Furthermore, it facilitates waste management prioritization, includes stakeholder participation, visualizes and, communicates results in an understandable manner to move towards CIWM. The i-ZEWATA is also less expensive and requires fewer resources to complete than an LCA. Thus, it is an appropriate and cost-effective methodology to apply in developing countries as an SCP tool to support CIWM in a circular economy and can additionally be applied to develop programs to reduce industrial waste's carbon footprint as part of CIWM.

The case study's results are limited. It was only applied to one case study. Additional studies should be done by applying the i-ZEWATA model to other case studies in the manufacturing and heavy industrial sectors. Both MCDA and VSM as a system engineering model and system assessment tool have their limitations and advantages; however, their applicability strongly depends on the context and required outcome influenced by data availability. Data availability is further influenced by the quality and the representativeness of the data. Industrial waste data is not always available in the iron and steel industry in developing countries due to challenges associated with inefficient and inaccurate waste recording, a lack of equipment to accurately weigh industrial waste, operational accountability challenges, a lack of trained personnel to manage waste systems and a lack of a centralized waste monitoring system.

Policymakers in developing countries may consider this research a starting point to address the limited available SCP tools to support the development of CIWM in the circular economy. The ZW model developed can be applied as an SCP tool in the circular economy to assess an industrial waste management system's current state in the iron and steel industry towards implementing a CIWM system. The ZW can also be used to assess the state of readiness of an iron and steel facility to participate in cleaner production, industrial symbiosis, and other circular economy initiatives. The ZW model can also be applied to conduct performance assessments towards ZW and CIWM system performance.

A future research direction should include demonstrating the ZW model's implementation with the i-ZEWATA methodology in other iron and steel facilities and other manufacturing sectors to develop and assess CIWM systems performance. There is a need to assess the readiness of priority industries to participate in circular economy initiatives such as industrial ecology programs. Another need is to include a bridging or CIWM development program to work with various industries to promote and integrate circular economy pathways in the form of CIWM in their current industrial waste management activities and systems. Even though debates still exist in avoiding rank reversal in AHP, future applications should also consider applying the COMET method combined with an AHP to address the rank reversal paradox in waste management decision-making challenges.

The proposed ZW model comprises a systems approach to promote CIWM in industrial waste management that supports an inclusive variety of environmental, technical, social, economic, regulatory, and innovative considerations. The ZW model enables waste and sustainability managers to prioritize ZW aspects associated with industrial waste management and valorization system in an iron and steel facility to move towards a circular economy. The ZW model also enables stakeholder and employee participation in developing and implementing a ZW industrial waste system. The ZW model is regarded as a holistic and system engineered model that can support public-private partnerships for practical cooperation between various stakeholders to promote sustainability benchmarks in developing countries in a circular economy. Further, the model contributes to iron and steel companies' resilience in developing countries and ultimately positions the industry towards a circular economy.

**Supplementary Materials:** The following are available online at https://www.mdpi.com/2071-1050/13/5/2832/s1, Table S1: Initial implementation and adaptive management of the ZW model using the i-ZEWATA methodology for the iron and Steel industry in developing countries; Table S2: Industrial waste facility or complex management and waste valorization plan components. Guidance on implementing the model and ZW model performance assessment and review following implementation is indicated in Table S1. The ZW model's initial implementation is suitable for industries that have not yet implemented the ZW model. The initial implementation process consists of seven steps (Table S1). Each step has been divided into the ZW model implementation framework components, input, suggested methods, components, tools, or data for the suggested input and the output. A ZW model blueprint or master plan needs to developed for facility complex management and valorization. This plan needs to be reviewed annually or when significant system and valorization changes are required. The facility or complex management plan needs to include both system and valorization management components. The proposed facility plan components are indicated in Table S2.

**Author Contributions:** Conceptualization, Y.S.; methodology, Y.S.; validation, Y.S.; investigation, Y.S.; resources, Y.S.; data curation, Y.S.; writing—original draft preparation, Y.S. and P.O.; writing— review and editing, Y.S. and P.O.; visualization, Y.S. and P.O.; supervision, P.O. and V.S.; project administration, Y.S. and P.O.; funding acquisition, Y.S. and P.O. All authors have read and agreed to the published version of the manuscript.

**Funding:** This research was funded by the University of the Free State with grant number 2004049256 and the Baoberry Centre of Innovation.

**Data Availability Statement:** The data supporting the reported results can be found at https://doi.org/10.6084/m9.figshare.13102538.v1.

**Acknowledgments:** The authors would like to thank the University of the Free State for the opportunity to complete the study, Marthie Kemp for her administrative support and also Lucas Mlangeni, Etienne Schoeman, Leon Bredenhann and Anré Weststrate for their support in this study.

**Conflicts of Interest:** The authors declare no conflict of interest.

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
