# Peer review of "A Zero-Waste Multi-Criteria Decision-Support Model for the Iron and Steel Industry in Developing Countries: A Case Study"

_sustainability, doi:10.3390/su13052832_

Round 1
Reviewer 1 Report
The iron and steel industry consumes vast quantities of energy and causes environmental degradation through greenhouse gas emissions and industrial waste generation and disposal. In the whole world, there is a need to manage complex industrial waste in the iron and steel industry, which requires a system engineering approach to zero waste management as informed by multi-criteria decision-making. This paper present a hybrid four-step multi-criteria decision-support model, the i-ZEWATA (Industrial Zero Waste Tiered Analysis). I-ZEWATA os presented as a road map for understanding, design, assess, and evaluate the iron and steel industrial waste systems with the ultimate objective of moving towards and achieving a zero-waste footprint. The presented results demonstrate that iron and steel waste can be identified, visualized, prioritized, and managed to promote zero-waste by applying a system engineered approach. The paper is well written scientific work. However, some shortcomings should be eliminated. The list of my comments is as follows:
1. The last paragraph of Introduction should presented the rest structure of the work.
2. Please emphases the contribution of this work in the Introduction section.
3. In introduction or in methods section the short review of MCDA method is needed. There should be shown that other methods was considered and the proposed was the best fitted. Justification is very important. Short review of MCDA methods also is very important, e.g., Are mcda methods benchmarkable? a comparative study of topsis, vikor, copras, and promethee ii methods; A new approach to identifying a multi-criteria decision model based on stochastic optimization techniques; Efficiency of methods for determining the relevance of criteria in sustainable transport problems: a comparative case study; or similar
4. Eq (1) should be written by using symbols
5. 2.1.2 AHP method. What dou you doing do prevent Rank Reversal? e.g. The rank reversals paradox in management decisions: The comparison of the ahp and comet methods or similar
6. Please extend in conclusions a future research direction
Author Response
Dear Reviewer 1,
Thank you for the valuable comments and suggestions and thank you for making time to review our manuscript. It is much appreciated.
The following changes were made based on your comments and suggestions received:
Reviewer 1 comments:
"The iron and steel industry consumes vast quantities of energy and causes environmental degradation through greenhouse gas emissions and industrial waste generation and disposal. In the whole world, there is a need to manage complex industrial waste in the iron and steel industry, which requires a system engineering approach to zero waste management as informed by multi-criteria decision-making. This paper present a hybrid four-step multi-criteria decision-support model, the i-ZEWATA (Industrial Zero Waste Tiered Analysis). I-ZEWATA os presented as a road map for understanding, design, assess, and evaluate the iron and steel industrial waste systems with the ultimate objective of moving towards and achieving a zero-waste footprint. The presented results demonstrate that iron and steel waste can be identified, visualized, prioritized, and managed to promote zero-waste by applying a system engineered approach. The paper is well written scientific work. However, some shortcomings should be eliminated. The list of my comments is as follows:
- The last paragraph of Introduction should presented the rest structure of the work.
Changes made:
A paragraph was inserted as the last paragraph in the Introduction section to present the manuscript's structure.
"The structure of the manuscript is as follows: first, the i-ZEWATA model and methodology are described. Second, the methodology as applied to the case study is illustrated, and the results presented. Finally, conclusions are drawn from the study's findings, with particular emphasis on the model as an SCP tool to promote CIWM.”
- Please emphases the contribution of this work in the Introduction section.
Changes made:
The contribution of the work was revisited and expanded on in the introduction section.
“In South Africa, which acts as a case study area for the current study, the existing industrial waste management approach is (1) capital intensive; (2) is indicative of in-adequately designed and implemented industrial waste management systems, and (3) waste treatment systems; (4) underlines a lack of understanding of the nature of industrial waste; (5) lacks appropriate methodology to guide industrial waste management and (6) treatment systems development; (7) implements, assesses and treats waste as a non-value adding component to the triple bottom line and is (8) limited in providing SCP tools to promote CIWM [48]. The reason for undertaking the current study was to develop a zero-waste (ZW) multi-criteria decision-support model for the iron and steel industry that will guide waste management, treatment, and valorization towards systems engineering and assessment, design, implementation, and evaluation. The developed ZW multi-criteria decision-support model aims to promote waste valorization as a value-adding component to the triple bottom line and guides the iron and steel industry towards developing a CIWM system, enabling a circular economy.
The study proposes a hybrid ZW multi-criteria decision support model, i-ZEWATA (Industrial Zero Waste Tiered Analysis), to develop a ZW model and implement a system engineered approach to waste management and treatment for the iron and steel industry with the ultimate objective of achieving ZW. According to the authors’ knowledge, this is the first study to develop a ZW multi-criteria decision-support model for the iron and steel industry for developing countries in Africa that can additionally be applied as an SCP tool to promote CIWM.”
- "In introduction or in methods section the short review of MCDA method is needed. There should be shown that other methods was considered and the proposed was the best fitted. Justification is very important. Short review of MCDA methods also is very important, e.g., Are mcda methods benchmarkable? a comparative study of topsis, vikor, copras, and promethee ii methods; A new approach to identifying a multi-criteria decision model based on stochastic optimization techniques; Efficiency of methods for determining the relevance of criteria in sustainable transport problems: a comparative case study; or similar."
Changes made:
Inserted the following under section 2.1:
“The most applied decision-making (DM) techniques as part of MCDM applied in waste management include complete aggregation models such as the “Technique for Order Preference with Similarity to Ideal Solution” (TOPSIS) and “Choosing by Advantages” (CBA). Partial aggregation techniques such as AHP, and Fuzzy Logic Decision Making [49] also comprise the most applied DM techniques [49,50]. Following the completion of an extensive literature review, AHP (including ANP and fuzzy AHP) was found to be the most dominant MCDA tool used, “Preference Ranking Organization Method for Enrichment Evaluations” (PROMETHEE) was an emerging method from the studies, “ELimination Et Choix Traduisant la REalité” (ELECTRE) was the most consistently used method whilst the diversity of MCDA methods was also expanding to include new methods [51]. AHP was found to be the most widely used and popular method to apply to a variety of problems that require complex decision-making [51] and for determining weights of criteria in MCDA [52,53]. The growth of publications related to the AHP method stands out above all other techniques [54]. The significance of AHP is based on the principle that to make a decision, people's knowledge and experience are considered as valuable as the data used [49]. It can deal with intangible and or quantifiable data and is carried out in two phases: hierarchy design and evaluation [49]. Experience and knowledge of the problem area are required to design the hierarchies [55].
The application of the ANP methodology is also documented widely, although only recently in the past two decades, with application in strategic policy planning [56-57]; in civil engineering [58]; territorial and environmental assessment [59-64]; and manufacturing systems [65,66]. The ANP, as a multi-criteria methodology, uses a com-plex model to consider a wide range of qualitative and quantitative criteria [67]. The ANP methodology addresses many decision problems that cannot be structured hierarchically, where many decision problems imply interactions and dependence between the highest and lowest elements [68]. In a hierarchy, the importance of the criteria causes the importance of alternatives, and the importance of the alternatives causes the importance of the criteria [68]. Therefore, by the generalized approach of the super-matrices introduced by the AHP, the ANP extends the applications of the AHP to the cases of interdependent relationships between the assessment elements [69] and can also address the complications associated with rank reversals. Even though rank reversals are as part of decision making as rank preservation, rank can be preserved by using the ideal mode in AHP in both relative and absolute measurement [70-72].
- Eq (1) should be written by using symbols
Changes made:
Eq (1) was written by using symbols as
WE(%) = PW / WMW
= PW / (WMW + PW) (1)
- 2.1.2 AHP method. What dou you doing do prevent Rank Reversal? e.g. The rank reversals paradox in management decisions: The comparison of the ahp and comet methods or similar
Changes made:
It was added in section 2.1.
“In a hierarchy, the importance of the criteria causes the importance of alternatives, and the importance of the alternatives causes the importance of the criteria [68]. Therefore, by the generalized approach of the super-matrices introduced by the AHP, the ANP extends the applications of the AHP to the cases of interdependent relationships between the assessment elements [69] and can also address the complications associated with rank reversals. Even though rank reversals are as part of decision making as rank preservation, rank can be preserved by using the ideal mode in AHP in both relative and absolute measurement [70-72].”
- Please extend in conclusions a future research direction
Changes made:
Additional part added to section 5.
“A future research direction should include demonstrating the ZW model's implementation with the i-ZEWATA methodology to other iron and steel facilities and other manufacturing sectors to develop and assess the performance of CIWM systems. There is a need to assess the readiness of priority industries to participate in circular economy initiatives such as industrial ecology programs. Another need is to include a bridging or CIWM development program to work with various industries to promote and integrate circular economy pathways in the form of CIWM in their current industrial waste management actions and systems. “
Reviewer 2 Report
The manuscript (sustainability-1099014) is well written and the author(s) have adapted the Multi-criteria-decision-analysis (MCDA) method to develop zero waste management strategies for the steel and iron sector for the case of South Africa. The content and method adaptation is interesting and novel and could be interesting to the wider audience working within the sustainability and circular economy paradigm. However, there are several shortcomings in the present form, mostly due to the presentation style and structure of the manuscript. Therefore, I would strongly advise author(s) to work on the following comments to improve the quality and clarity of the manuscript.
1) Systems engineering is used in this paper, so it is then essential to present the diagram of the system showing all the process flow and sub-processes. This diagram is typically a system life cycle or process flow diagram of the product. Authors should include a brief process flow diagram of the iron/steel industry showing what are the inputs and waste streams that they want to target. This baseline diagram would then form a basis when authors discuss the zero-waste strategies. This could be included at the beginning of Section 3 (for example).
2) Figure 1 and section 4 is positioned very late in the manuscript. This should be positioned as soon as the method section (section2) to guide the readers on what each step represents and how the logical flow is organized in the ZW model. In the current manuscript, it is increasingly confusing for readers to understand the logical flow of each step. Authors can refer to the previous MCDA based studies to see how the logical flow or algorithm of the steps should be entered in the manuscript at the very beginning.
3) In Section 4 should be made to discuss the findings in detail. The study findings have policy relevance. So how ZW model results identify areas of improvement should be stressed. The links to sustainability and circular economy should be highlighted here effectively than in the current state.
4) Finally, all the MCDA studies are based on some subjectivity and associated limitations. These should be explicitly stated in the limitations section in section 5. The authors do mention the limitation of the method to only the steel/iron sector, but other methodological limitations, data limitations should be elaborated further.
Author Response
Dear Reviewer 2,
We would like to sincerely thank you for making time to review our manuscript. Thank you for the kind comments and also for the very insightful suggestions. We have made amendments to the manuscript following your comments and suggestions.
Reviewer 2 comments
The manuscript (sustainability-1099014) is well written and the author(s) have adapted the Multi-criteria-decision-analysis (MCDA) method to develop zero waste management strategies for the steel and iron sector for the case of South Africa. The content and method adaptation is interesting and novel and could be interesting to the wider audience working within the sustainability and circular economy paradigm. However, there are several shortcomings in the present form, mostly due to the presentation style and structure of the manuscript. Therefore, I would strongly advise author(s) to work on the following comments to improve the quality and clarity of the manuscript.
1. Systems engineering is used in this paper, so it is then essential to present the diagram of the system showing all the process flow and sub-processes. This diagram is typically a system life cycle or process flow diagram of the product. Authors should include a brief process flow diagram of the iron/steel industry showing what are the inputs and waste streams that they want to target. This baseline diagram would then form a basis when authors discuss the zero-waste strategies. This could be included at the beginning of Section 3 (for example).
Changes made:
In agreement with the suggestion. The outcome of the actual VSM map can be used to indicate the process flow diagram of the iron and steel waste and hence can be used as a baseline diagram as aligned to the main objectives of the manuscript. The actual VSM map was inserted at the beginning of section 3 as Figure 2.
2. Figure 1 and section 4 is positioned very late in the manuscript. This should be positioned as soon as the method section (section2) to guide the readers on what each step represents and how the logical flow is organized in the ZW model. In the current manuscript, it is increasingly confusing for readers to understand the logical flow of each step. Authors can refer to the previous MCDA based studies to see how the logical flow or algorithm of the steps should be entered in the manuscript at the very beginning.
Changes made:
An overview of the i-ZEWATA model, as Figure 1, was developed and subsequently included in section 2 to allow for a better understanding of the logical flow of the thought and development process regarding the model. However, we have left the main discussion around the i-ZEWATA model as Figure 2 in section 4 because it is the primary outcome of the manuscript.
3. In Section 4 should be made to discuss the findings in detail. The study findings have policy relevance. So how ZW model results identify areas of improvement should be stressed. The links to sustainability and circular economy should be highlighted here effectively than in the current state.
Changes made:
The detailed findings are discussed in section 3, where the method was applied to the case study. The findings are discussed in detail pertaining to the outcome of the VSM and the combined AHP and ANP process in developing a multi-criteria decision-support (and not decision-making) model. However, it is possible that the model can be applied in decision-making with consequent policy relevance. However, that is outside the scope of this particular manuscript.
Also added to section 4:
" The ZW system engineered model's goal functions as a CIWM SCP tool that aims to promote a thorough analysis of industrial waste management to assist in the decision-support process of developing and implementing CIWM systems in the circular economy. In developing countries, the ZW model can address uncertainty in complex industrial waste systems in the iron and steel industry while promoting the United Nation’s Sustainable Development Goal 12 as SCP. The ZW model couples a system engineering model with system assessment tools in a hybrid configuration. The model also addresses the variable of waste composition and becomes a precursor to developing an optimal, adaptive and resilient CIWM system for the iron and steel industry in developing countries. The ZW model can also be applied in the design process and assessment of CIWM from a systems perspective. The outcome of implementing the ZW model through the i-ZEWATA methodology can deliver on creating waste exchange opportunities in industrial symbiosis and waste internalization specifically applicable to legacy manufacturing facilities where waste avoidance and minimization are not always feasible due to aging manufacturing technology and facility layouts. Also, the ZW model, through promoting decision-support, can promote full CIWM accounting by connecting horizontal and vertical subprocesses to determine the connections between waste generation and waste treatment and therefore not only prepare industrial facilities to participate in industrial symbiosis programs but also to measure the readiness of a particular facility to participate in such programs.
Guidance on implementing the model and ZW model performance assessment and review following implementation are indicated in Annexure A1. The ZW model performance assessment and review can be conducted annually to review the ZW model regarding social, technological, legal, economic, environmental applicability, practicality, and progress. The ZW model performance assessment and review can only be done following the ZW model's initial implementation or following the implementation of existing ZW initiatives. “
Additional section added to section 5
" Policymakers in developing countries may consider this research a starting point to address the limited available SCP tools to support the development of CIWM in the circular economy. The ZW model described can be applied as an SCP tool in the circular economy to assess an industrial waste management system's current state in the iron and steel industry towards implementing a CIWM system. The ZW can also be used to assess the state of readiness of an iron and steel facility to participate in cleaner production, industrial symbiosis, and other circular economy initiatives. The ZW model can also be applied to conduct performance assessments towards ZW and CIWM system performance."
4. Finally, all the MCDA studies are based on some subjectivity and associated limitations. These should be explicitly stated in the limitations section in section 5. The authors do mention the limitation of the method to only the steel/iron sector, but other methodological limitations, data limitations should be elaborated further.
Changes made:
Section 5 was expanded to include the suggestions.
"Both MCDA and VSM as a system engineering model and system assessment tool have their limitations and advantages; however, their applicability strongly depends on the context and required outcome influenced by data availability. Data availability is further influenced by the quality and the representativeness of the data. Industrial waste data is not always available in the iron and steel industry in developing countries due to challenges associated with inefficient and inaccurate waste recording, a lack of equipment to accurately weigh industrial waste, operational accountability challenges, a lack of trained personnel to manage waste systems and a lack of a centralized waste monitoring system."
Round 2
Reviewer 1 Report
In my opinion, the authors have made good progress. However, it is not yet sufficient to accept the work. Please pay particular attention to points 3 and 5 of the previous review. Some of the relevant literature has been omitted. Once again I would like to stress the necessity to indicate the background of the MCDA methods as VIKOR method etc. At the same time showing that the methods give different results is strongly recommended (detailed hints in the previous review), e.g., in the paper: 'Do distance-based multi-criteria decision analysis methods create similar rankings?' authors have shown that the different results are given. Then, the justification of using the proposed approach will be more clear. In conclusion, I suggest to fill in the gaps.
Author Response
Dear Reviewer 1,
Thank you again very much for your comments. It is much appreciated.
Several changes were made to the manuscript following Round 2 review. We include the details of the changes below.
Responses to the comments received by Reviewer 1.
Round 2 comments
"In my opinion, the authors have made good progress. However, it is not yet sufficient to accept the work. Please pay particular attention to points 3 and 5 of the previous review. Some of the relevant literature has been omitted. Once again I would like to stress the necessity to indicate the background of the MCDA methods as VIKOR method etc. At the same time showing that the methods give different results is strongly recommended (detailed hints in the previous review), e.g., in the paper: 'Do distance-based multi-criteria decision analysis methods create similar rankings?' authors have shown that the different results are given. Then, the justification of using the proposed approach will be more clear. In conclusion, I suggest to fill in the gaps."
Round 1 comments
"3. In introduction or in methods section the short review of MCDA method is needed. There should be shown that other methods was considered and the proposed was the best fitted. Justification is very important. Short review of MCDA methods also is very important, e.g., Are mcda methods benchmarkable? a comparative study of topsis, vikor, copras, and promethee ii methods; A new approach to identifying a multi-criteria decision model based on stochastic optimization techniques; Efficiency of methods for determining the relevance of criteria in sustainable transport problems: a comparative case study; or similar
- 2.1.2 AHP method. What dou you doing do prevent Rank Reversal? e.g. The rank reversals paradox in management decisions: The comparison of the ahp and comet methods or similar"
Response
Thank you very much for your insightful comments. We have updated the methodology section to:
- Include the relevant literature as suggested,
- Expanded on the background information and review section of Multi-Criteria Decision Analysis methods,
- Indicated that the methods provide different results and hence strengthened the justification of the approach,
- Expanded on rank reversal prevention in the methodology section and included a future research direction in the conclusion.
Changes made based on reviewer comments received:
Changes made:
Line 157-161: Although there are a large number of MCDA methods available, no particular method is perfect for applying in every decision-making situation to solve every decision problem [49-50]. Differences in decision recommendations and different results can be an outcome when applying different MCDA methods to decision-making challenges [50-52]. When addressing a specific decision-making challenge, selecting the MCDA method needs to be appropriate to the specific problem that needs to be solved [50,53].
Line 168: added "…in waste management."
Line 174-183: "The application of the MCDA methods "VlseKriterijumska Optimizacija I Kompromisno Resenje" (VIKOR) [50,57] and "Complex Proportional Assessment" (COPRAS) in waste management was found not to be as extensively applied in waste management challenges as AHP. A possible explanation relates to the proper determination of criteria weights and how the various MCDA methods are applied (whether as a singular or as combination method) that affects the final ranking [50,58,59]. As a new method, the characteristic objects method (COMET) [60,61] also has merit in the future application in waste management due to factors associated with the ease of identifying non-linear and linear decision-making and the independence of assessed alternative sets to the assessed decision variants [60,61]."
Line 203-207: "Even though rank reversals are as part of decision making as rank preservation, rank can be preserved by using the ideal mode in AHP in both relative and absolute measurement [80-82] and possible future integration of the COMET method and supporting methodology needs to be investigated [83]."
Line 932-935: "Even though debates still exist in avoiding rank reversal in AHP, future applications should consider applying the COMET method in combination with an AHP to address the rank reversal paradox in waste management decision-making challenges."
We have also updated the reference list to include additional references as used in the methodology section. Reference numbering in-text was adapted.
Please do not hesitate to contact me if you require any additional information or clarification. I look forward to hearing from you in due course.
Yours faithfully,
Yolandi Schoeman
Round 3
Reviewer 1 Report
I suggest accepting in the current form.
Author Response
Dear Reviewer 1,
Thank you for your much appreciated assistance in helping us to improve the manuscript. Thank you for suggesting that the journal accepts our manuscript in the current form.
Kind regards,
Yolandi Schoeman